

# Effects of food type and abundance on begging and sharing in Asian small-clawed otters (*Aonyx cinereus*)

Madison Bowden-Parry, Erik Postma and Neeltje J. Boogert

Department of Biosciences, University of Exeter, Penryn, Cornwall, United Kingdom

## ABSTRACT

Begging for food, a conspicuous solicitation display, is common in a variety of taxa, and it has received extensive research attention in a parent-offspring context. Both theoretical models and empirical evidence suggest that offspring begging can be an honest signal of hunger or a mediator of competition between siblings. At a behavioural mechanistic level, begging for food can be a form of harassment aimed at persuading those in possession of food to share. Food sharing, defined as the transfer of a defendable food item from one individual to another, can vary considerably between species, age-classes and food type and abundance. We investigated the determinants of begging and food-sharing behaviours in Asian small-clawed otters (*Aonyx cinereus*), a group-living species that commonly exhibits begging in captivity. We presented two captive otter populations with three food types that varied in exploitation complexity, in three different abundances. We predicted that begging rates would be highest when food was in lowest abundance and hardest to exploit, and that increased begging would lead to increased food sharing. We found that, over time, increased begging rates were indeed correlated with increased food transfers, but neither food type complexity nor abundance affected begging or sharing rates. However, age category was significantly associated with begging and food sharing rates: juvenile otters begged more and shared less than adult otters. The results from this first experimental study on begging and food sharing within the Mustelid family begin to reveal some of the drivers of these behaviours.

## INTRODUCTION

Begging for food, a conspicuous display that can involve both physical and vocal solicitation behaviours (*Kilner & Johnstone, 1997*; *Wright & Leonard, 2007*), is widespread across taxa (*Von Bayern et al., 2007*; *Carter & Wilkinson, 2013*; *Gilby, 2006*; *Goodall, 1986*; *Hauser, 1992*; *Hauser & Marler, 1993*; *Silk et al., 2013*) and has received extensive research attention in a parent-offspring context (*Trivers, 1974*; *Royle, Hartley & Parker, 2002*). Theoretical models of begging have focussed on the nutritional demands of offspring and food solicitation interactions with their parents (*Grafen, 1990*; *Zahavi, 1975*). These models, and empirical evidence, support two main explanations for begging: an 'honest signal of hunger' (*Bowers et al., 2019*; *Kilner & Johnstone, 1997*; *Godfray, 1991*; *Godfray, 1995*; *Mock*

Corresponding author
Madison Bowden-Parry, madison-bowdenparry@gmail.com

& *Parker, 1997*), which is most commonly tested in young birds (e.g., *Capodeanu-Nägler et al., 2017*; *Christe, Richner & Oppliger, 1996*; *Karasov & Wright, 2002*), and competition between siblings for the acquisition of food from parents, known as the 'sibling scramble hypothesis' (*Harper, 1986*; *Stamps, Metcalf & Krishnan, 1978*; *Macnair & Parker, 1979*; *Parker, 1985*). As begging and the resultant food sharing are typically documented to occur between kin, these interactions can be explained by inclusive fitness benefits (*Hamilton, 1964*).

Although food sharing is common between relatives, it also occurs between unrelated conspecifics, mates, adults and non-biological offspring (*De Kort, Emery & Clayton, 2006*; *Feistner & McGrew, 1989*; *Jaeggi, Van Noordwijk & Van Schaik, 2008*). Food sharing is commonly observed in nonhuman animals, including primates (*De Waal, 2000*; *Jaeggi & Van Schaik, 2011*; *Silk et al., 2013*), corvids (termed 'allofeeding', *De Kort, Emery & Clayton, 2006*; *Heinrich, 1988*; *Stacey & Koenig, 1990*; *Thiollay, 1991*), leaf-nosed bats (vampire bats: *Carter & Wilkinson, 2013*; *DeNault & McFarlane, 1995*; *Wilkinson, 1984*), oceanic dolphins (*Fedorowicz, Beard & Connor, 2003*; *Hoelzel, 1991*; *Wright et al., 2016*), rodents (spiny mice: *Porter, Moore & White, 1981*) and big cats (*Cooper, 1991*). It is widely defined as the transfer of a defendable food item from one food-motivated individual to another, which in many circumstances is an unresisted transfer (*De Kort, Emery & Clayton, 2006*; *Feistner & McGrew, 1989*; *Hadjichrysanthou & Broom, 2012*). However, the exact definition of food sharing appears to depend on species and context (*Hadjichrysanthou & Broom, 2012*).

While food sharing between kin is easily explained through inclusive fitness benefits, at a mechanistic level food sharing may be motivated by harassment avoidance (*Carter & Wilkinson, 2013*; *Gilby, 2006*; *Jones, 1984*; *Stevens & Stephens, 2002*; *Wrangham, 1975*; *Wright & Leonard, 2007*). The harassment avoidance (or 'sharing-under-pressure') hypothesis states that persistent begging might coerce conspecifics into sharing their food. Similar to the 'tolerated theft' model of food sharing in humans (*Jones, 1984*; *Blurton Jones, 1987*; *Isaac, 1978*), sharing can provide benefits to food owners. By sharing, food owners can reduce further harassment by beggars, and thus gain the benefits of dismissing the recipient and eating the remainder of their food in peace (*Jones, 1984*; *De Kort, Emery & Clayton, 2006*; *Stevens & Stephens, 2002*; *Stevens, 2004*; *Wrangham, 1975*). However, unlike tolerated theft, the harassment avoidance hypothesis in not contingent upon hunger asymmetry (*Jones, 1984*; *Blurton Jones, 1987*; *Isaac, 1978*) and predicts that (1) harassment is costly to the food owner, (2) harassment elicits food sharing and (3) food sharing reduces harassment (*Stevens & Stephens, 2002*). In this study, we tested whether begging and sharing frequencies were correlated within two family groups.

Both begging and food sharing frequencies vary considerably between species, age groups and contexts (*Boesch, Hohmann & Marchant, 2002*; *Hadjichrysanthou & Broom, 2012*). Factors such as food availability and food type may explain some of this variation (*Elgar, 1986*). For example, food availability has been shown to influence offspring begging frequencies in various bird species (*Quillfeldt & Masello, 2004*; *Price & Ydenberg, 1995*; *Smith & Montgomerie, 1991*), with food limitation increasing overall begging activity and intensity (*Budden & Wright, 2001*; *Kilner & Johnstone, 1997*; *Watson & Ritchison, 2018*). Food type can also influence begging frequency: increased begging for novel food types

has been observed in several primate species, such as chimpanzees (*Nishida & Turner, 1996*), marmosets (*Brown, Almond & Bates, 2005*), lion tamarins (*Price & Fiestner, 1993*) and cotton-top tamarins (*Feistner & Chamove, 1986*), although this is not always the case (bonobo; *Kuroda, 1984*). Studies of how both food abundance and food type influence begging and food sharing frequencies in non-primate mammals remain rare. Here, we tested whether some of the variation in begging and sharing frequencies may be explained by food abundance and food type in two family groups of otters (*De Waal, 2000*).

Food sharing by adults with dependent offspring is common among various primate species (*Feistner & Price, 1990*; *Feistner & Price, 2000*; *Hiraiwa-Hasegawa, 1990*; *Jaeggi, Van Noordwijk & Van Schaik, 2008*; *Nishida & Turner, 1996*; *Rapaport & Ruiz-Miranda, 2002*; *Ruiz-Miranda et al., 1999*; *Silk, 1978*; *Ueno & Matsuzawa, 2004*). In some species of Callitrichids (*Feistner & Price, 1990*; *Feistner & Price, 2000*; *Rapaport & Ruiz-Miranda, 2002*; *Ruiz-Miranda et al., 1999*), food transfers mostly involve 'difficult-to-process' foods, are controlled by adults, and peak during weaning. However, in chimpanzees, most food transfers of difficult-to-process foods occur before weaning (*Hiraiwa-Hasegawa, 1990*; *Nishida & Turner, 1996*; *Silk, 1978*). In some cases, such food transfers are begged for and controlled by juveniles rather than adults (*Jaeggi, Van Noordwijk & Van Schaik, 2008*; *Nishida & Turner, 1996*; *Ueno & Matsuzawa, 2004*). In orangutans, offspring solicitation is dependent upon foraging skill level, with more competent offspring soliciting fewer food items than dependent offspring (*Jaeggi, Van Noordwijk & Van Schaik, 2008*). In this study, we tested whether otter age classes differed in their foraging efficiency and begging and sharing frequencies.

We studied the gregarious Asian small-clawed otter, the smallest of all 13 extant species within the Mustelid family (*Aziz, 2018*; *Foster-Turley, 1992*; *Hussain, Gupta & de Silva, 2011*; *Wayre, 1978*). This species is a suitable model system to examine drivers of food sharing behaviour, because they form large family romps (ca. 12 individuals), exhibit extreme begging behaviour in captivity (i.e., incessant, loud vocalisations towards zookeepers and each other prior to and during provisioning times) that can be quantified to test for harassment avoidance, and they are abundant in zoos and wildlife centres. Asian small-clawed otters feed on both fish and invertebrates in the wild (*Chanin, 1985*; *Mason & Macdonald, 1987*). As a hand-oriented species, juveniles are thought to be independent foragers at 13 months old (*Watt, 1991*; *Watt, 1993*) and have a very high metabolic rate (*Marshall, Lekagul & McNeely, 1988*). Some individuals will consume 20–25% of their body mass in food per day (*Costa & Kooyman, 1982*; *Kenyon, 1969*; *Morrison, Rosenmann & Estes, 1974*), suggesting that the energetic cost of food sharing, as well as begging, might be relatively high in this species.

We presented two captive family groups with three food types that varied in the complexity of their prey defences (inferred by exploitation latency) in three different abundances. First, we tested the prediction from the harassment avoidance hypothesis (*Stevens & Stephens, 2002*; *Wrangham, 1975*) that, assuming harassment was costly to food owners, otters would show more food sharing the more begging they received. We also predicted that, given their high metabolic rate and nutritional needs, otters would beg more for food when food abundance was low, and when food was difficult to exploit. However,

in these circumstances (i.e., of low food abundance), one might expect less sharing as food owners might be less responsive to the begging of group mates when their own nutritional needs take priority. We also predicted that juveniles would harass (i.e., beg) more and share less than adults, given that juvenile animals typically show less efficient foraging behaviour, and are nutritionally dependent on adults (*Bolnick et al., 2003*; *Capodeanu-Nägler et al., 2017*).

## MATERIALS & METHODS

### Ethical statement
The University of Exeter departmental ethics committee (2018/2334), Newquay Zoo and the Tamar Otter and Wildlife Centre approved all experiments in this study.

### Subjects
This study was conducted on two groups of otter, all of which were captive-born and raised. The group kept at the Tamar Otter and Wildlife Centre in Launceston consisted of 12 related subjects including 6 adults and 6 juveniles, while the group kept at Newquay Zoo in Newquay consisted of 8 related subjects, of which 3 were adults and 5 juveniles. All juveniles were 3 to 12 months old (see Table S1). Otters were identified by distinctive markings on their face and body. The groups differed in their regular feeding regime; at Tamar, food was provided at variable times of day and their diet (standardised in weight) consisted of various food items, including whiting, clams, oysters and fish-based biscuits. Occasionally the keepers would introduce chicks, helmeted guinea fowl and queen scallops. The feeding regime at Newquay followed a strict schedule, with the group being fed at 9 AM, 12 AM, 4 PM and 6 PM, and their diet consisted of chicks, mice, minced beef meatballs and hard-boiled eggs. The observations and experiments described below took place from March to May 2018.

### Experimental food presentations
To investigate how food sharing and harassment would vary with food-extraction complexity and food abundance, we conducted 9 experimental food presentations per group. The two otter groups were tested within their normal zoo enclosures. Prior to experimental trials, otters in Newquay were fed ∼14–20 chicks for their 9 AM feed, and 8 mice, 6 hard-boiled eggs and 8 minced beef balls for their midday feed. The Tamar group varied in their feeding times throughout the day. However, they were always fed whiting and fish biscuits prior to the experimental treatments. The experimental treatments took place at least one hour after the otter's regular feeding times and were undertaken between 12-4 PM. This limited any food-driven anticipatory behaviour and fullness from a previous feed.

We varied food complexity by presenting three natural prey types: rainbow trout (*Oncorhynchus mykiss*), shore crab (*Carcinus maenas*) and blue mussels (*Mytilus edulis*), which we presented on consecutive days in a randomised order that was kept the same for both otter groups. Shore crabs averaged eight cm in diameter and the rainbow trout were cut into five ca. 3 cm-width pieces. These food types were novel to the Newquay group and

represented a non-regular prey type to the Tamar group. The hard shells of shore crabs and mussels required dexterous manipulation by otters to exploit, whereas the soft flesh and lack of protective shell made trout an easily exploitable food type. We inferred food type complexity by the amount of time otters spent to exploit each food item, measured for five adults and five juveniles in each otter group (i.e., $n = 20$ observations per food type) from first touch to consumption of the piece of food. This confirmed that trout was the easiest (i.e., fastest) to exploit (average latency $\pm$ SD (seconds): $80.83 \pm 32.58$ s), followed by mussels ($112.75 \pm 45.46$ s), with crabs taking the longest to consume ($224.96 \pm 93.55$ s). The abundance of each prey type was manipulated to be either half as many food items as there were otters (low), the same number of food items as there were otters (medium) or double the number of food items as there were otters (high).

Each otter group experienced one trial per day, starting with the most complex of prey, then medium, then easy to exploit, presented in the medium, then low, then high abundance all 9 trials (representing all possible combinations of food type and abundance) were completed. Due to only two otter families being studied, food type was not randomised across groups to avoid confounding group ID and food type order. In between trials, there was an average inter-trial interval of 2 days. This interval was implemented to reduce anticipatory behaviour towards the trials, which could have influenced the otters' behaviours. For each trial, the zookeepers would close the otters inside their pen whilst we placed the food at random locations within the same 3 $\times$ 3 m vicinity inside the otter enclosure for each treatment. Once the otters were released, we filmed the group until all food items were consumed. We filmed all trials with three video cameras placed such that they covered the whole enclosure area.

### Video analyses

We scored all videos of experimental food presentations using the definitions adapted from *De Waal (1993)* and *De Waal (1997b)*. A 'beggar' was defined as any individual that approached within an 'otter's arm's length' of the 'food owner', while making vocal solicitations.

### Food sharing behaviours

(i) collect near: beggar retrieves food item from near owner's arms, or within an otter's arm length of owner. There is no retaliation from the food owner when food item is taken;

(ii) relaxed claim: beggar takes part of the food, or the whole food item that the food owner possesses (i.e., in the owner's paws) without resistance from owner;

(iii) food giving: owner facilitates transfer of food by actively moving it toward another otter.

### Harassment behaviours

(iii) passive begging: beggar sits near and stares at the food owner (no physical contact), within ca. $\leq 1$ m of food owner. The beggar will often be in the owner's view, but the beggar can sit and stare from behind, commonly whilst making long, repetitive vocalisations directed at the food owner.

(iv) active begging: beggar physically contacts owner or food. The owner has full view of the beggar, and the beggar's paws are either on the food item itself, or on the food owner's paws. Active begging can be distinguished from an attack or other aggressive interactions as the latter involved lunges with or without physical contact (e.g., grabbing the other's side of their neck with their teeth). Aggressive interactions or attacks were exceedingly rare and usually lasted no longer than 10 s.

We scored the number of sharing and harassment behaviours for each individual in each food presentation trial video. The food sharing behavioural category 'collect near' (e.g., collecting food scraps or broken off pieces within an otter's arm's length of a feeding individual) was included in the analysis as a voluntary food transfer; in each instance when an otter was successful, and collected a piece of food near the owner, there was clear possession of food by the owner. On occasions when 'collect near' was unsuccessful, i.e., the approaching otter tried but failed to collect a nearby piece of food, the food owner responded by either collecting the nearby piece themselves and turning away, turning their head to stare at the approaching otter, or engaging in a lunge towards the approaching otter. This suggested that, by reacting to the approaching otter's presence, the food owner regarded the nearby food item as within their possession.

This categorisation of food sharing and begging behaviours is commonly used to examine food interactions and strategies (*Boesch & Boesch, 1989*; *De Waal, 1989*; *De Waal, 1997a*; *De Waal, 1997b*; *Stevens, 2004*; *Watt, 1993*). We also recorded any harassment behaviours that resulted in food transfers and the IDs of the initiator and receiver of food transfers. Coding of the videos started when the first piece of food was touched by an individual and ended when all food items were eaten or after 45 minutes of food exposure.

## Statistical analyses

All analyses of begging and sharing behaviours were performed in R 3.6.1 (*R Development Core Team, 2019*) using generalised linear mixed-effect models (GLMMs) implemented in the lme4 package (*Bates et al., 2014*). The total number of active and passive begging occurrences (the most common type of harassment behaviours observed), and the sum of all types of voluntary sharing behaviours (i.e., collect near, relaxed claim and food giving), per otter and trial duration were modelled assuming a Poisson error distribution and using a log link function. Trial duration measured in seconds was log-transformed and included as an offset term (offset(log(trialduration/60))) to account for the variable lengths of each food presentation trial, effectively transforming the number of begging and sharing events into a begging and sharing rate (the number of events per minute). To account for overdispersion, we included an observation-level random effect (OLRE) (*Harrison, 2014*). Furthermore, to control for repeated measures of the same 20 individuals, and for the fact that we recorded the behaviour of all individuals within each of the 18 experimental trials (9 per zoo), we included both otter ID and trial number as random effects in our model structure. To test for the overall significance of our key predictors, we compared the full models with respective null models using a Likelihood ratio test (hereafter LRT), using the LRT R function "drop1" with argument 'test' set to 'Chi' (*Bates et al., 2014*; *Dobson, 2002*).

Firstly, to test if otters were more likely to share if they received more begging, which would support the harassment avoidance hypothesis, we constructed a GLMM with Poisson error distribution using the total number of 'food transfers' (i.e., both voluntarily and following harassment) for each food presentation trial for each otter as our response variable. We fitted the total number of instances of 'begging received' per otter per trial, and the offset for trial duration, as our only fixed effects. We included the observation-level random effect as well as otter ID and trial as random effects.

Before testing whether food type, food abundance, otter sex and age affected begging frequency, we first checked whether juvenile otters were less efficient foragers than adults, and whether any difference in foraging efficiency between age categories increased with food type complexity. Such age differences in foraging efficiency might then explain age differences in begging frequencies. We analysed the five (randomly selected) observations of food exploitation latency per otter age category per food type per zoo with a linear mixed-effects model (LMM). The response variable was the log-transformed latency to exploit the food type from first touch to complete ingestion, while the fixed effects were food type, and age category. Zoo ID was included as a random effect.

We then constructed a Poisson-GLMM with the total number of active and passive begging occurrences per food presentation trial per otter as the response variable. Food type (crabs, mussels or trout), food abundance (low, medium or high), as well as sex, otter age category (juvenile or adult), zoo and the offset for trial duration were included as fixed effects, and we included the observation-level identifier, otter ID and trial as random effects. To improve convergence, we used the "BOBYQA" optimizer. To test whether the same predictors affected food sharing frequencies, we ran the exact same model, but using the total number of sharing occurrences per food presentation trial per otter as the response variable.

To test if the age composition of food owner-receiver dyads influenced the frequency of voluntary sharing, we fitted another Poisson-GLMM using the number of directional voluntary sharing occurrences summed for each potential dyad in each otter group across all trials and treatments as our response variable. We fitted the age composition of the otter dyad (i.e., Adult sharing with Adult, Juvenile-Juvenile, Adult-Juvenile or Juvenile-Adult) and the offset for trial duration as our fixed effects. We included the observation-level random effect as well as the ID of the sharing individual and the ID of the receiving individual as our random effects. We also performed the post-hoc Tukey multiple comparison test to compare the different age composition-dyads for significant differences in directional sharing frequencies.

To detect and identify any outliers within our models, we used an exploratory analysis approach where we calculated the interquartile range for our non-normally distributed response residuals and the upper and lower Tukey fences (upper fence; Q3 + 1.5 * IQR, lower fence = Q1 −1.5 * IQR; *Tukey, 1977*) to determine any data points outside of this range. Of 20 subjects, there were 12 individuals within our 'total number of begging occurrences' variable that were considered to be outliers. These 12 individuals were all juveniles (≤12 months). Our 'total number of sharing occurrences' revealed 1 individual to be an outlier, our 'total number of food transfers' revealed no outliers and our 'total

times that begging was received' contained 11 individuals classed as outliers. The extreme values, or 'outliers' in our 'total number of begging occurrences' were all generated by juveniles and the extreme values in our 'total times that begging was received' were mostly adults being begged to. This indicated that there were age group differences and thus, all 'outliers' were included in our analyses (code available on Figshare and see Tables S2–S4 for results excluding 'outliers').

## RESULTS

### Is begging correlated with food transfers?

There was a positive relationship between the frequency with which otters received begging and the number of times they transferred food: the more begging individuals received, the more often they transferred food (Table 1, Fig. 1; LRT for begging received: $X^2$ (1) 13.19, $P = < .001$; 95% confidence interval [CI] 0.04–0.14). Effect size estimates showed that for every begging bout, otters were estimated to receive on average 1.6 food transfers (ranging from 0.55–3.31 for the shortest and longest trial durations, respectively).

### Do food type and abundance affect begging and sharing frequencies?

Juveniles took significantly longer than adults to exploit the experimentally provided foods (LRT for age category: $X^2(1) = 9.75$, $P = 0.002$). However, this difference in foraging efficiency between juveniles and adults did not increase with food type complexity (LRT for food type-age interaction: $X^2(2) = 0.40$, $P = 0.82$). Across age categories, food type did not significantly affect otters' begging rates (LRT for food type: $X^2$ (2) $= 3.60$, $P = 0.17$; 95% confidence interval food type trout CI $[-1.20–0.22]$; 95% confidence interval food type mussel CI $[-1.37–0.05]$. Although otters appeared to beg less when more food was available (Table 2, Fig. 2), food abundance did not significantly affect otters' begging rates either (LRT for food abundance: $X^2$ (2) $= 4.09$, $P = 0.13$; 95% confidence interval high food abundance CI $[-1.40–0.05]$; 95% confidence interval medium food abundance CI $[-0.76–0.64]$).

Neither food abundance nor food type affected otters' sharing rates (Table 3; LRT for food abundance: $X^2$ (2) $= 1.79$ $P = 0.41$; 95% confidence interval high food abundance CI $[-0.33–0.86]$; 95% confidence interval medium food abundance CI $[-0.21–0.99]$). LRT food type: $X^2$ (2) $= 0.92$, $P = 0.63$; 95% confidence interval food type trout CI $[-0.56–0.64]$; 95% confidence interval food type mussel CI $[-0.78–0.38]$; Table 3).

### Do otter age, zoo and sex correlate with begging and sharing frequencies?

Juveniles begged significantly more than adults over all trials (Table 2, Fig. 3; LRT for age category: $X^2$ (1) $= 7.53$, $P = 0.006$; 95% confidence interval juveniles CI [0.21–1.10]). There were no significant effects of otter sex or zoo on begging frequencies (LRT for zoo: $X^2$ (1) $=2.67$, $P = 0.10$, sex: $X^2$ (1) $= 0.04$, $P = 0.84$; 95% confidence interval Tamar group CI $[-0.12–1.17]$; 95% confidence interval sex: male CI $[-0.40–0.48]$). Effect size estimates showed that for each time an adult begged, juveniles begged 2.19 times (ranging from 0.73–4.40 for shortest and longest trial durations). We found that juveniles shared

**Table 1** Generalised linear mixed effects model showing how the amount of begging an otter received influenced how often they shared food. Intercept refers to total "sharing" when begging received is 0. The significant values are highlighted in bold.

| GLMM output Begging and food transfers | | B | SE | df | z | p |
|---|---|---|---|---|---|---|
| **Fixed effects** | | | | | | |
| (Intercept) | | −2.655 | 0.146 | | −18.226 | <.001 |
| Begging received | | 0.091 | 0.025 | 1 | 3.562 | **<.001** |
| **Random effects** | | Variance | | | | |
| obs | (intercept) | 0.233 | | | | |
| ID | (intercept) | 0.067 | | | | |
| trial | (intercept) | 0.144 | | | | |
| Observation | 180 | | | | | |
| ID | 20 | | | | | |

significantly less than adults over all trials (Table 3, Fig. 4; LRT for age category; $X^2$ (1) = 12.85, $P = < .001$; 95% confidence interval juveniles [CI] −1.29- −0.43). An overall effect size estimate of 0.77 showed that juveniles shared ca. 23% less than adults (ranging from 0.26–1.55 for shortest and longest trial durations, respectively). There was no significant effect of zoo or otter sex on sharing frequency (LRT for zoo: $X^2$ (1) = 0.39, $P = 0.53$, sex: $X^2$ (1) = 0.53, $P = 0.47$; 95% confidence interval Tamar group CI [−0.75–0.39]; 95% confidence interval sex: male CI [−0.57–0.29]).

The age composition of otter dyads had a significant effect on sharing (LRT for sharing dyad age composition: $X^2$ (3) = 13.179, $P = 0.004$); juvenile dyads (J-J) shared significantly less than adult dyads (A-A) (Tukey post-hoc test, $P = 0.034$) and juveniles shared significantly less with adults than adults did with each other (Tukey post-hoc test, $P = < .001$). All other comparisons of age composition were not significant (>0.05) (see Table 4, Fig. 5).

## DISCUSSION

### Harassment avoidance or 'sharing-under-pressure'

Otters showed both passive and active begging during our experimental trials, and food owners responded to both forms of harassment (i.e., behaviours that restrict the food owner's ability to eat their food in peace) by sharing their food. This suggests that harassment is costly to the food owner, and that this cost forces them to share instead. Our results thus provide support for the harassment avoidance/sharing-under-pressure hypothesis, which assumes begging imposes high energetic and opportunity costs on the food owner (Stevens & Stephens, 2002; Wrangham, 1975). While harassment hence seems to be a parsimonious explanation for the food sharing observed in our captive otter family groups, the drivers of food sharing in this species could be further explored. For example, one could experimentally vary the ease with which food items can be monopolised, as well as quantify the relative costs of begging and sharing food.

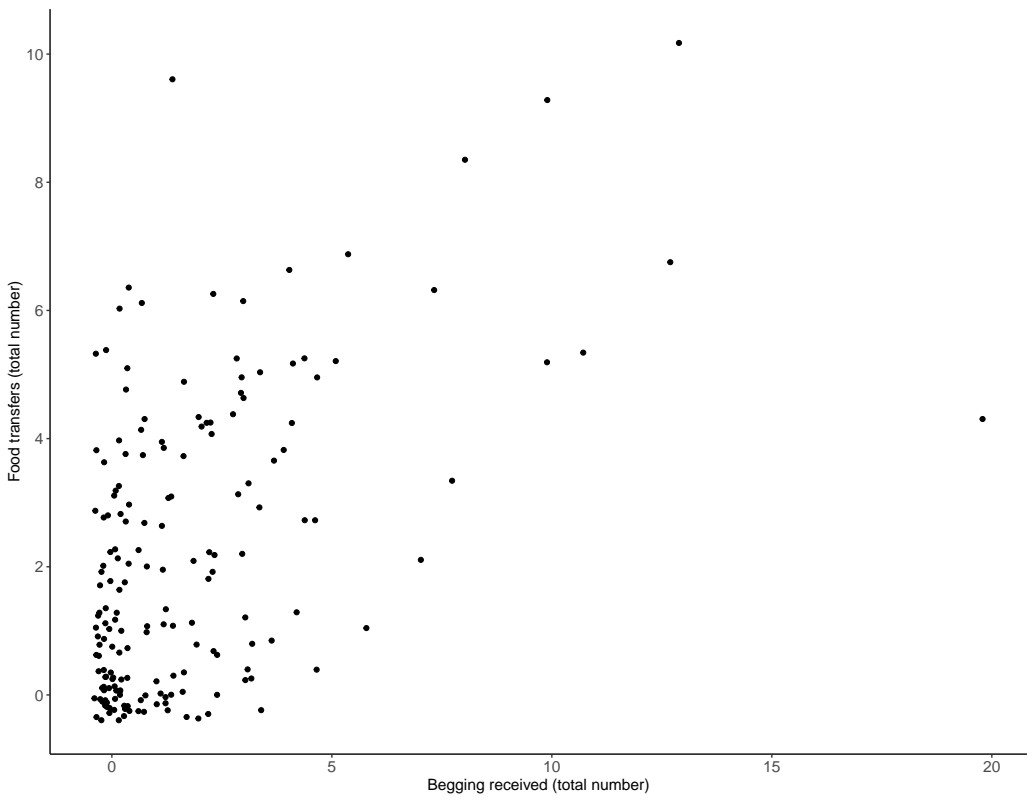

**Figure 1    Number of individual food transfers across all trials, as a function of begging received.** The more begging individuals received, the more often they transferred food. Points indicate individual otters. Jitter has been applied to data points to visualise those lying on top of each other.

## Begging behaviour and food type/abundance treatments

Although begging was somewhat less frequent when food was abundant, neither food type nor abundance had a significant effect on begging or sharing frequencies. Captive otters often show incessant begging, which has been proposed to be related to excessive hunger (*Gothard, 2007*) and/or boredom (*Hawke et al., 2000*). Excessive hunger seems an unlikely cause of the otters' begging behaviour, given that zoos follow guidelines of adequate nutrition, diets and feeding regimes, and part of the otters' daily diet is often used as enrichment to encourage natural foraging skills and reduce boredom (*IUCN/SSC Otter Specialist Group, 2012*). However, it should also be noted that the exact nutrient requirements for captive otters are still unknown and these guidelines can vary dramatically between species and institutions (*IUCN/SSC Otter Specialist Group, 2012*; *Lombardi, 2002*).

Wild otters have a high metabolic rate and spend up to 60% of their time foraging (*Davis et al., 1992*; *Marshall, Lekagul & McNeely, 1988*; *Spelman, 1999*; *Hoover & Tyler, 1986*). Their diet is nutrient-dense, and wild otters eat small amounts and frequently (*IUCN/SSC Otter Specialist Group, 2012*; *Marshall, Lekagul & McNeely, 1988*). For a species with such a high energy requirement and an opportunistic foraging strategy, we expected to see an increase in begging behaviour during our low food abundance treatment. Although

**Table 2  Generalised linear mixed effects model results showing how food type, food abundance, age category, sex and zoo correlated with the total amount of begging shown by otters.** Intercept refers to "crab" food type, "low" food abundance, "adult" age category, "female" sex and "Newquay" zoo. The significant values are highlighted in bold.

| GLMM output Begging | | B | SE | df | z | p |
|---|---|---|---|---|---|---|
| Fixed effects | | | | | | |
| (Intercept) | | −2.935 | 0.395 | | −7.429 | <.001 |
| Food type | crab | 0 | 0 | | | |
| | mussel | −0.625 | 0.335 | 2 | −1.869 | 0.062 |
| | trout | −0.490 | 0.339 | 2 | −1.444 | 0.149 |
| Food density | low | 0 | 0 | | | |
| | medium | −0.078 | 0.331 | 2 | −0.236 | 0.813 |
| | high | −0.673 | 0.343 | 2 | −1.966 | **0.049** |
| Age category | adult | 0 | 0 | | | |
| | juvenile | 0.655 | 0.218 | 1 | 3.004 | **0.003** |
| Sex | female | 0 | | | | |
| | male | 0.043 | 0.215 | 1 | 0.201 | 0.840 |
| Zoo | Newquay | 0 | 0 | | | |
| | Tamar | 0.529 | 0.310 | 1 | 1.706 | 0.088 |
| Random effects | | Variance | | | | |
| obs | (Intercept) | 0.433 | | | | |
| ID | (Intercept) | 0.081 | | | | |
| trial | (Intercept) | 0.198 | | | | |
| Observation | | 180 | | | | |
| ID | | 20 | | | | |

food limitation can increase begging frequencies in some food-sharing species (*Quillfeldt & Masello, 2004*; *Price & Ydenberg, 1995*; *Smith & Montgomerie, 1991*), occurrences of begging in other species appear to be independent of food availability and ownership (*Kuroda, 1984*). Here, we observed otters begging across all food abundance conditions and regardless of food ownership.

Our experimental manipulation of food type also did not appear to affect begging or sharing frequencies, which might be due the fact that most otter species are generalists (*Timm-Davis, DeWitt & Marshall, 2015*), and their diets reflect the local abundance of food (*Almeida et al., 2012*; *Clavero, Prenda & Delibes, 2003*; *Krpo-Ćetković et al., 2019*; *Lanszki & Körmendi, 1996*). Asian small-clawed otters are manual predators who rely on extractive foraging, using their clawless fingers to reach into crevices and sieve through silt (*Ladds, Hoppitt & Boogert, 2017*). Their jaws are well equipped to crack hard shells compared to other mouth-oriented species and their partially webbed paws allow for greater dexterity (*Sivasothi & Nor, 1994*; *Timm-Davis, DeWitt & Marshall, 2015*). Although all of the food types used in this study represented novel prey to the captive otters we tested, they are part of a typical diet for wild Asian-small clawed otters (*Chanin, 1985*; *Mason & Macdonald, 1987*). This adaptation to a generalist diet (*Timm-Davis, DeWitt & Marshall, 2015*) could partly explain why food type had no effect on begging frequency in this study.

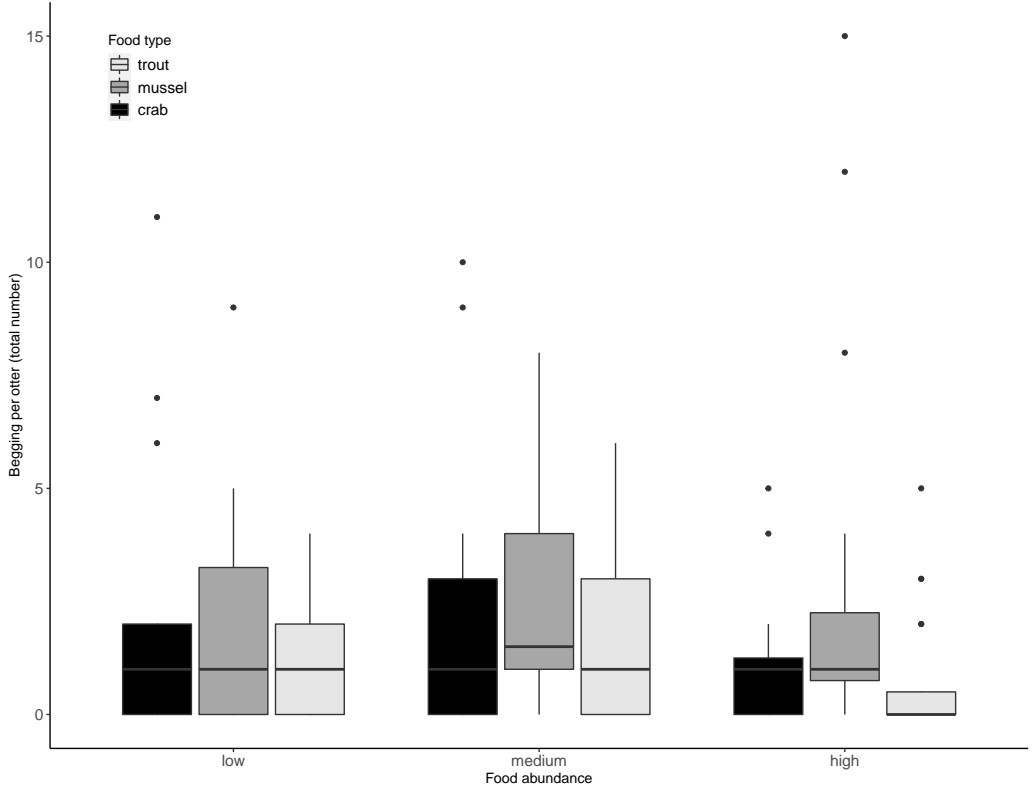

**Figure 2** **Individual begging occurrences across all trials, as a function of food abundance and type.** Begging increased when less food was available, but not significantly so. Boxes represent the interquartile range and the bars within the boxes are the median values. Whiskers indicate values within ±1.5 times the interquartile range.

## Begging behaviour and otter age

Juveniles exhibited more begging behaviour than adults, and within the group of juveniles, the youngest individuals begged more than others. Explanations for juveniles begging for food have centred around begging being an honest signal of hunger (*Bowers et al., 2019*; *Kilner & Johnstone, 1997*; *Godfray, 1991*; *Mock & Parker, 1997*) and competition between siblings (*Harper, 1986*; *Stamps, Metcalf & Krishnan, 1978*; *Macnair & Parker, 1979*; *Parker, 1985*). According to the honest signalling hypothesis (*Bowers et al., 2019*; *Kilner & Johnstone, 1997*; *Godfray, 1991*; *Mock & Parker, 1997*), the costly qualities of extravagant begging behaviours advertise an offspring's current nutritional need (*Cotton, Kacelnik & Wright, 1996*; *Kilner, 1995*; *Kilner & Johnstone, 1997*; *Ottoson, 1997*). However, this idea remains somewhat controversial (*Mock, Dugas & Strickler, 2011*; *Parker, Royle & Hartley, 2002*), and others have argued that stronger, rather than hungrier, individuals can maintain higher and longer levels of begging (*Grafen, 1990*; *Parker, Royle & Hartley, 2002*).

In captive Asian small-clawed otters, behaviours such as food handling, chewing and swallowing are energetically expensive (*Borgwardt & Culik, 1999*), and in some otter species, juveniles have similar mass-specific energy demands as adults (sea otters: *Thometz et al., 2014*). In addition to needing food constantly, foraging skills in some species of otters do
**Table 3 Generalised linear mixed effects model showing how food type, food abundance, age category, sex and zoo correlated with the total amount of voluntary food sharing by otters.** Intercept refers to "crab" food type, "low" food abundance, "adult" age category, "female" sex and "Newquay" zoo. The significant values are highlighted in bold.

| GLMM output Sharing | | B | SE | df | z | p |
|---|---|---|---|---|---|---|
| Fixed effects | | | | | | |
| (Intercept) | | −2.477 | 0.355 | | −6.983 | <.001 |
| Food type | crab | 0 | 0 | | | |
| | mussel | −0.195 | 0.280 | 2 | −0.697 | 0.486 |
| | trout | 0.060 | 0.292 | 2 | 0.207 | 0.836 |
| Food density | low | 0 | 0 | | | |
| | medium | 0.395 | 0.289 | 2 | 1.365 | 0.172 |
| | high | 0.253 | 0.290 | 2 | 0.872 | 0.383 |
| Age category | adult | 0 | 0 | | | |
| | juvenile | −0.844 | 0.206 | 1 | −4.105 | **<.001** |
| Sex | female | 0 | | | | |
| | male | −0.155 | 0.208 | 1 | −0.742 | 0.458 |
| Zoo | Newquay | 0 | 0 | | | |
| | Tamar | −0.173 | 0.276 | 1 | −0.625 | 0.532 |
| Random effects | | Variance | | | | |
| obs | (Intercept) | 0.157 | | | | |
| ID | (Intercept) | 0.084 | | | | |
| trial | (Intercept) | 0.127 | | | | |
| Observation | 180 | | | | | |
| ID | 20 | | | | | |

**Table 4 Generalised linear mixed effects model showing how the age composition of otter dyads correlated with the total amount of voluntary food sharing between them.** Intercept refers to "adult-adult" sharing. The first individual indicates the food sharer while the second indicates the receiver. Significant values are highlighted in bold.

| GLMM output Directional sharing | | B | SE | df | z | p |
|---|---|---|---|---|---|---|
| Fixed effects | | | | | | |
| (Intercept) | | −4.505 | 0.2109 | | −21.357 | <.001 |
| Age category | AA | 0 | 0 | | | |
| | AJ | −0.393 | 0.216 | 3 | −1.821 | 0.069 |
| | JA | −1.037 | 0.272 | 3 | −3.811 | **<.001** |
| | JJ | −0.794 | 0.295 | 3 | −2.694 | **0.007** |
| Random effects | | Variance | | | | |
| obs | (Intercept) | 0.142 | | | | |
| Breceive | (Intercept) | 0.092 | | | | |
| Ashare | (Intercept) | 0.172 | | | | |
| Observation | 188 | | | | | |
| ID | 20 | | | | | |

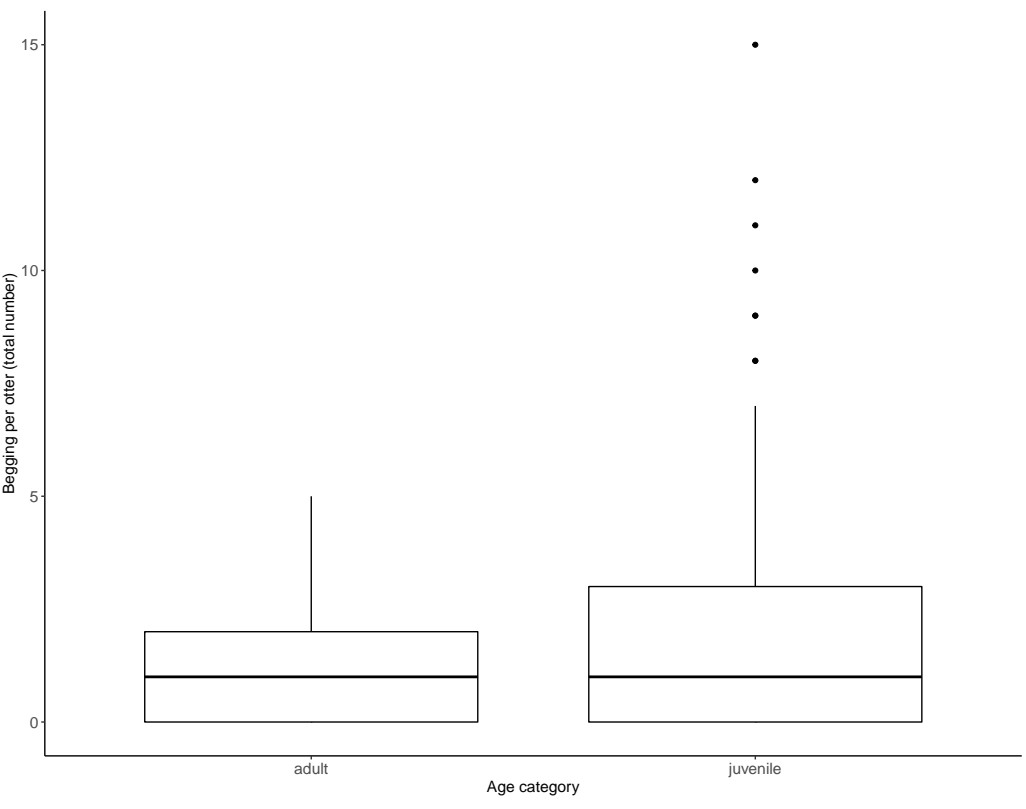

**Figure 3  Individual begging occurrences across all trials, as a function of age category.** Juveniles begged more than adults. Boxes represent the interquartile range and the bars within the boxes are the median values. Whiskers indicate values within ±1.5 times the interquartile range.

not fully develop until 13 months old (*Watt, 1991*; *Watt, 1993*). In our study, the most frequently begging individuals were ∼3–12 months old. Perhaps, the extensive begging displayed by these young individuals was indeed an honest signal of hunger, reinforced by competition with older siblings (*Field et al., 2005*). Moreover, the higher begging frequencies in these young individuals could also be influenced by foraging incompetence, and thus their dependence on older offspring and adults for the acquisition of food (*Jaeggi, Van Noordwijk & Van Schaik, 2008*), as we found that juveniles were less efficient foragers than adults across all food types. This difference in foraging competence might also help explain why we observed less food sharing between juveniles and from juveniles to adults as compared to adult dyads (see below).

## Food sharing behaviour, food type/abundance treatments and the effect of age

Food sharing frequencies have been observed to differ between food types in some species, including chimpanzees (*Boesch & Boesch, 1989*; *Rose, 1997*) and capuchins (*De Waal, 2000*), but not in others (*Kuroda, 1984*; bonobo). Although food abundance and quality thus have the potential to influence some food sharing instances in nature (*Elgar, 1986*), we did not find an effect of food abundance nor food type on sharing frequency. Our study
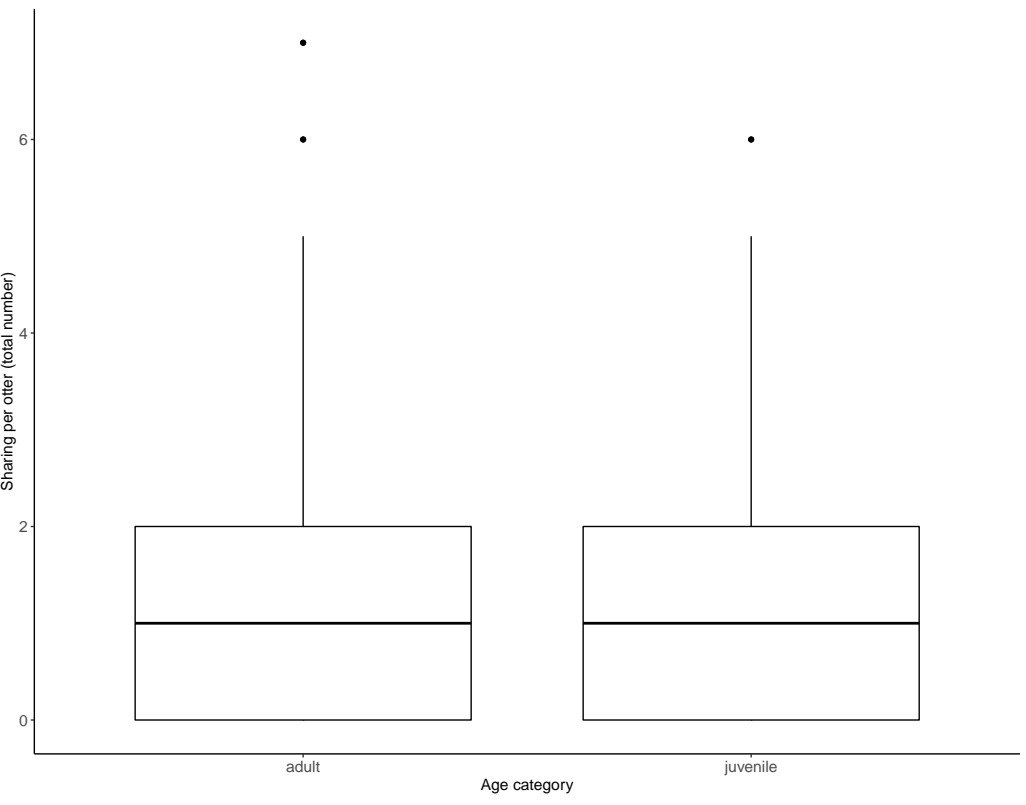

**Figure 4 Individual sharing occurrences across all trials, as a function of age category.** Adults shared more often than did juveniles. Boxes represent the interquartile range and the bars within the boxes are the median values. Whiskers indicate values within ±1.5 times the interquartile range.

suggests that these ecological factors appear to have little influence on food sharing in Asian small-clawed otters. This may be due to the fact that we studied otters in captivity; given that captive animals get provisioned several times per day, our food manipulations may have been less relevant and effective than if we had been able to study more food-motivated individuals in the wild.

Although food abundance and type did not influence sharing frequencies, we found that juvenile otters shared food significantly less often than did adults. Older animals tend to be more skilled and efficient at catching prey compared to juveniles (*Goss-Custard & Durell, 1987*; *Stalmaster & Gessaman, 1984*; *Woo et al., 2008*) and the inefficiency of juveniles' foraging behaviour can lead to intra-specific competition with older individuals (*Field et al., 2005*). In sea otters, juveniles are proficient at tool use and prey handling by 5 months of age, while juveniles in other hand-oriented otter species are proficient only at 13 months of age (*Payne & Jameson, 1984*; *Watt, 1991*; *Watt, 1993*). It is possible that the youngest individuals in our study were in competition with adults and older siblings for the acquisition of food, and were thus less inclined to share food, while the highly competent adult foragers could afford to share. It would be of value for future studies to examine age-class influences on food competition and food transfer behaviours in more detail.

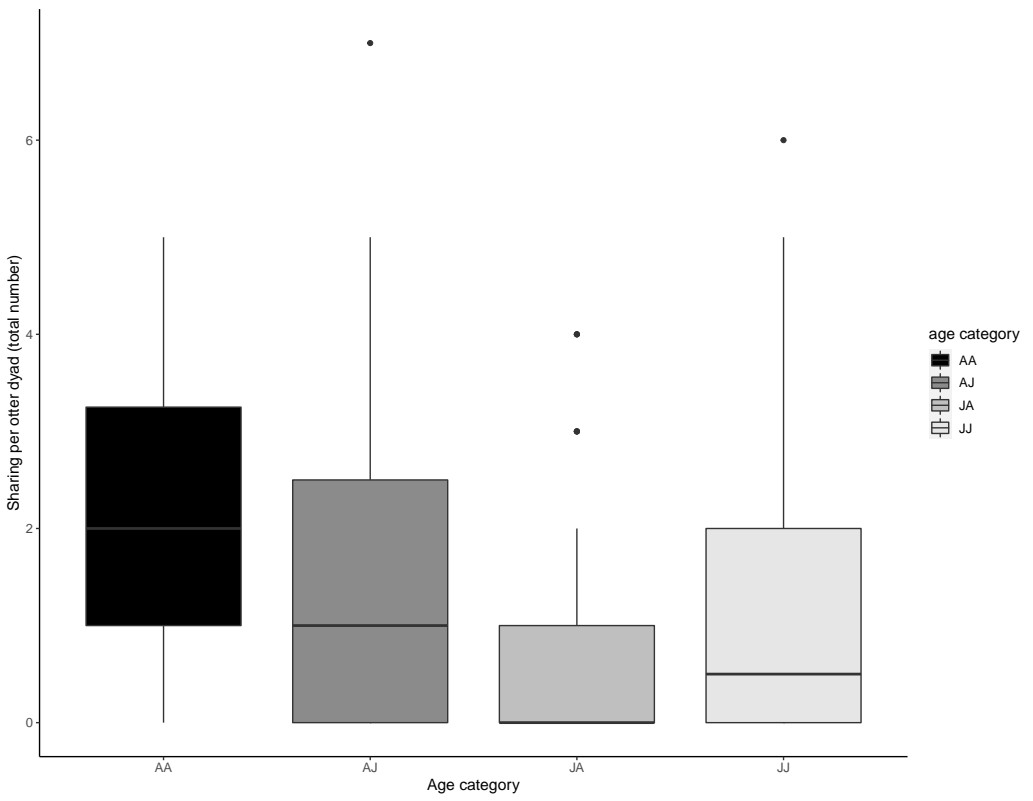

**Figure 5 Dyadic voluntary sharing occurrences across all experimental food presentations as a function of the otter dyad's age composition.** The first individual indicates the food sharer while the second indicates the receiver. Sharing was most frequent amongst adult dyads. Boxes represent the interquartile range and the bars within the boxes are the median values. Whiskers indicate values within ± 1.5 times the interquartile range.

## CONCLUSION

The experiments presented in this study provide the first exploration and insights into food sharing and the use of begging behaviours in the Asian small-clawed otter. We found that otters are more likely to share food with increased harassment, in the form of begging behaviours, supporting the harassment avoidance/sharing-under-pressure hypothesis. Furthermore, we found evidence that juveniles shared less and begged more than adults, with the youngest juveniles begging more than the others. However, food abundance and food type appear to have little influence on food sharing and begging behaviours in Asian small-clawed otters. It is important to note that this may be due to the fact that our subjects were in a captive setting, and thus our food manipulations may be less effective than they might be in wild and more food-motivated individuals. This study had a relatively small sample size and it is essential to repeat this study on more groups of otters with a larger range of ages and group sizes. The frequent begging behaviours observed during this study, and the considerable variation in begging across age groups, also warrant further study. In addition, we suggest that further investigation of food sharing in Asian small-clawed

otters considers the influence of social bonds on food transfers, as social dominance, partner-choice and social network structure might help further explain food sharing in this highly sociable species.

## ACKNOWLEDGEMENTS

We thank Newquay Zoo and Tamar Otter and Wildlife Centre for access to their otter groups. We also thank The University of Exeter for the loan of equipment and Georgina Hume, Tiffany Volle and Alex Saliveros for their field assistance.

### Funding

The authors received no funding for this work.

### Competing Interests

The authors declare there are no competing interests.

### Author Contributions

- Madison Bowden-Parry conceived and designed the experiments, performed the experiments, analyzed the data, prepared figures and/or tables, authored or reviewed drafts of the paper, and approved the final draft.
- Erik Postma analyzed the data, authored or reviewed drafts of the paper, and approved the final draft.
- Neeltje Boogert conceived and designed the experiments, analyzed the data, authored or reviewed drafts of the paper, and approved the final draft.

### Animal Ethics

The following information was supplied relating to ethical approvals (i.e., approving body and any reference numbers):

The University of Exeter's Biosciences Ethics committee provided full approval for this research.

### Data Availability

Data and code are available at Figshare: https://figshare.com/authors/Madison_Bowden-Parry/9424769.

Bowden-Parry, Madison; Postma, Erik; J. Boogert, Neeltje (2020): Bowden-ParryOtterdata.csv. figshare. Dataset. https://doi.org/10.6084/m9.figshare.13017158.v2

Bowden-Parry, Madison; Postma, Erik; J. Boogert, Neeltje (2020): Bowden-ParryOtterRScript.R. figshare. Software. https://doi.org/10.6084/m9.figshare.13016888.v2

Bowden-Parry, Madison; Postma, Erik; J. Boogert, Neeltje (2020): Bowden-ParryOtterDyadShare.csv. figshare. Dataset. https://doi.org/10.6084/m9.figshare.13017137.v1.
## Supplemental Information

Supplemental information for this article can be found online at http://dx.doi.org/10.7717/peerj.10369#supplemental-information.

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
