# Peer review of "Effects of food type and abundance on begging and sharing in Asian small-clawed otters (Aonyx cinereus)"

_PeerJ, doi:10.7717/peerj.10369_

## Round 0.1 · original submission · Minor Revisions

I was very fortunate to receive three very detailed and helpful reviews from experts on this topic and/or species. I was pleased to see a paper on begging and food sharing in a less commonly studied species and I agree with the reviewers that your data will make a nice contribution to the literature. I also agree that the paper is well written, and the rationale is nicely situated in current theory. However, I also agree with the reviewers that you need to do more work to clarify the terminology in your MS along with the details of the begging behavior itself along the lines suggested by the reviewers. I think you need to reconcile the predictions that animals would beg more under low abundance conditions with the prediction that animals would share more with more begging, because it also seems reasonable to assume that there would be less sharing (or tolerated theft, as the case may be) when food is scarcer. It seems then that the predictions are in conflict. You need to say more about extraction time as a measure of complexity in the introduction so that it is not introduced out of nowhere in the Methods. It needs to be clearer what the otters are fed around the trials (at the other feeding times that day). You need to be cautious in generalizing from these conditions in a captive setting where diet is very much controlled to a wild context where animals are uncertain as to their next food source. You do mention this in the discussion, but some more emphasis on this as a limitation of the study might be warranted. You will also need to address the relatively few observations (only one trial per condition in two families). It is not clear to me how your analysis accounted for who was doing the begging and who was being begged to, or who was doing the sharing (also reflected in the reviewers’ comments) and to whom, and how you accounted for the lack of independence of the data. Is the individual or the trial the unit of analysis?

Reviewer 1 ·

Basic reporting

Great. No comment.

Experimental design

No comment (but see below).

Validity of the findings

Good, no comment.

Additional comments

This is an interesting study on food sharing and begging in captive otters. The paper is well-written and well-organized. I have no major concerns with the methods or results, but I have some suggestions for improving the reporting of results. I have two major comments and minor comments by line below. Also, you’ll want to move you data and R code from dropbox to a repository like Figshare or Data Dryad or include it as a supplement.

Major comments

The definitions of food sharing and begging need to be clarified (in abstract, introduction and methods). How does one distinguish between begging and dominance behavior? As it is written, if one otter aggressively attacks another otter and takes its food that would count as “food sharing: collect near” because “beggar retrieves food from within arm’s reach of owner”. I would call this a ‘food transfer’ but definitely not sharing. Also active begging is defined as one otter “physically contacts owner or food”. So again, an attack would be considered begging. I don’t think these extremely broad descriptions (any physical contact) are the definitions the observers actually used (implicitly or explicitly) to define begging. For example, if one otters tail touches another is that begging? Or if one otter pushes another aside is that begging? Use definitions that would allow the study to be replicated properly. Or alternatively explain what these interactions typically look like so the reader can understand that they are actually ‘begging’ behaviors. A video would be helpful.

On lines 315 and 317 The authors say that the intensively begging individuals which were removed as outliers were the youngest individuals. If this is true, then these data are not outliers, and they should not be excluded from any analyses (and it also suggests a possible non-linear effect of age on begging frequency which makes sense). It would be good to plot begging frequency with age to visualize that.

Minor comments by line

Abstract
32 might be good to define begging somewhere
35 “(inclusive)” does not need to be in parentheses
42 “begging rates were indeed correlated with increased food transfers” Over time or across dyads
51 delete “both models of”
60 no parentheses needed
76 It seems strange to lump all insects together as one example, and use spotted hyenas as another. Rather than discussing food sharing in all animals, you could just focus more on a certain group, like on food-sharing in group-living mammals, and/or consider differences between cooperative breeders and non-cooperative breeders.
85 When you say “in species such as…” what do chimps, capuchins, and jackdaws have in common?
93 “(inclusive) fitness benefois” should say “indirect fitness benefits”. Also instances do not require an adaptive explanation, only heritable traits do.
94 “who are then expected to” should say “that are more likely to”
98 two commas missing
123 if ‘begging’ is in quotes, you should explain what this behavior looks like.

Methods
174 Can you explain why prey complexity was not randomised? Or clarify if it was or was not.
184 Do you mean a ‘possible beggar’? I don’t think begging should be defined purely based on proximity, and this also contradicts definitions below.
194 How do you discriminate active begging from fighting over food
205 total number of begging occurrences for the otter that is begging or being begged?
234 12 outliers out of how many total data? Same with the others…

Results
242 It would be good to report an effect size (standardized or unstandardized). What proportion of begging observations led to a food transfer (e.g. 12 out of 100)?
255 This section would also benefit from effect size estimates as in “juveniles begged XXX more than adults” or “shared XXX less than adults”.
263 change “significant trend” to “trend”. Also, you don’t need to say “however” because there is no real difference between p =0.04 and p=0.05.
for any nonsignificant results, I would suggest plotting the coefficient with 95% CI as a way to assess statistical power. I gave an example below.

Figure 1 needs to show the number of overlapping data points using transparency, data point size, or geom_jitter()

Figure 2 and 3 log-transformation, log(X+1), would show the data better

Discussion
315 - 317 see main comments


# R code for getting 95% confidence interval of a linear mixed model coefficient using bootstrapping

library(lme4)

# create fake data
data <-
data.frame(
response= round(runif(n=1000, min=0, max=10)),
predictor=sample(c(T,F), size=200, replace=T),
id= sample(c("a","b","c", 'd'), size=200, replace=T))


# fit model
fit <- summary(lmer(response ~ predictor + (1|id), data= data))

# get fixed effect coefficient
obs <- fit$coefficients[2,1]

# choose number of samples
perms <- 1000

# get expected coefficients from bootstrapping (might take awhile)
exp <- rep(NA, perms)
for (i in 1:perms){

# resample data with replacement
data2 <- data[sample(nrow(data), nrow(data), replace=T), ]

# fit model
fit2 <- summary(lmer(response ~ predictor + (1|id), data= data2))

# get fixed effect coefficient
exp[i] <- fit2$coefficients[2,1]
}

# plot bootstrapped values
hist(exp)

# plot confidence interval of coefficient
library(ggplot2)
ci95 <- quantile(exp, probs= c(0.025, 0.975))
d <- data.frame(low= ci95[1], mean= obs, high=ci95[2])
ggplot(data= d, aes(x="", y=mean))+
geom_point(size=3)+
geom_errorbar(aes(ymin=low, ymax=high, width=.1), size=1)+
geom_hline(yintercept = 0)+
ylab("coefficient")+
xlab("")+
ggtitle(“bootstrapped 95% confidence interval”)

Reviewer 2 ·

Basic reporting

This is an interesting article reporting food-sharing experiments among captive Asian small-clawed otters. The experiments are framed in terms of literature addressing the function and variation of begging as well as sharing behaviors. I found the literature to be quite comprehensively reviewed, though I have some suggestions for increasing clarity (see below). The figures and tables are appropriate and the statistical inferences are fine. I congratulate the authors on this nice study.

My major suggestion is that the introduction could be clearer. There are a lot of issues brought up, and I think a better structure would help readers not to lose track. I think in some cases the logic of a certain argument (e.g. why some food types are shared more than others) could also be spelled out more. Essentially, the article has to introduce functional explanations of begging behavior, sharing behavior, and factors influencing variation in these behaviors (such as food type and abundance). In addition, the authors also appear to distinguish between begging/sharing among relatives and among non-relatives. This is a lot to keep track of, and not all nuances are fleshed out. For example, what is arguably more important than the distinction between relatives and non-relatives is whether begging/sharing occurs between adults and juveniles, who are not yet competent foragers, or among adults, who are competent foragers. For a juvenile, begging may be the only way to obtain food even when it's abundant, if it cannot yet independently process the food, hence some studies distinguish between "difficult" and "easy" foods (Silk 1978, Nishida & Turner 1996, Jaeggi et al 2008). Here, the authors inferred extraction complexity from exploitation time, but it is unclear how this extraction complexity varied by age - were all otters able to process all food types with similar efficiency or did juveniles struggle with e.g. crabs and were better off begging from more competent adults? - and whether begging and sharing rates were explained by foraging competence. Among adults, who are all competent foragers, what matters is monopolizability - again, the question is can an individual obtain food independently (by simply walking over to a different part of the enclosure) or is all the food controlled by a single individual and everyone else's only chance of obtaining anything is to beg? So here the most important property of a food type is how large/clumped it is - e.g. in chimpanzees meat is shared much more than fruit because it comes in a single, large package whereas fruit is all around. It seems like in the present study the monopolization potential was relatively low for all food types (they all came in smallish pieces), but varied by abundance (a small pile is more easily monopolized than a large pile). On the other hand, some explanations like reciprocity or social bonding (which may arguably be the same thing) don't really need to be introduced since the authors don't go on to test whether sharing is reciprocated or relates to the relationship between owner and beggar in any other way. In fact, given that begging is the most important predictor of sharing and that you're not taking into account any relationships, it seems somewhat redundant to test predictors of begging (table 2) and sharing (table 3) - juveniles beg more (table 2), and begging predicts sharing (table 1) hence juveniles are shared with more (table 3); this is not that surprising (though still interesting and important to report!). What is surprising is the relative lack of a food type or abundance effect, which is why it's important to clearly contextualize this in terms of the relevant factors of processing difficulty and monopolizability - some of this is more explicitly mentioned in the Discussion, and some better streamlining between Introduction and Discussion would therefore be helpful.

In sum, I would urge the authors to rephrase some of the introduction, especially when talking about variation in begging/sharing due to food properties in terms of extraction difficulty (for juveniles) and monopolizability (for adults). The order of the introduction also appears slightly jumbled, with e.g. the first paragraph introducing begging, the second variation in begging and sharing, and then the third seems to go back in defining sharing. The "between relatives" on line 111 also seems out of place as previous paragraphs already talked about sharing among non-relatives. In sum, a lot of the relevant literature and factors are discussed, but appear slightly jumbled and not always clearly fleshed out. I think some targeted revisions would make the paper much clearer.

Experimental design

The authors provided the otters with several different food types in different abundances and recorded any begging and sharing behaviors. The respective behaviors are well defined and quantified. The statistical methods are well explained and appropriate.

I have a few minor questions regarding begging and sharing behavior. Passive begging: were food possessors ever disturbed by this and did it ever lead to food sharing without subsequent active begging? In my study species, I have never seen an owner care about passive begging. Similarly, did owners ever attempt to resist "collect near" or appear upset by it in any way? Again, I have never seen owners bothered by this, and it led me to conclude that they must not regard the food lying near them as being in their possession and therefore lay no claim to it. For this reason, "collect near" does not seem to fit the definition of sharing (the food is arguably not being transferred because it is not in possession of anyone) and several studies exclude it. In your case, if you have reason to believe that collect near does constitute a transfer, i.e. that owners claim possession over food lying near them and at least sometimes resist attempts to take it, then there is no need to change, but otherwise it might be useful to re-run your analyses excluding collect near.

I would also urge the authors to follow the recent open science movement and make their data and code publicly available for easier replicability.

Validity of the findings

No further comments

Additional comments

Minor comments by line numbers:

50-52: duplicated sentence

87: Feistner, not Fiestner

100-110: should also credit Blurton Jones 1984, who coined the term tolerated theft. He extended the logic of the Hawk-Dove game to situations where food is clumped and comes in large packages. In this case, beggars have no other choice but to try to obtain food from owners, and should be highly motivated to do so (i.e. play Hawk, if needed), because they won't eat otherwise. Owners on the other hand will be less motivated to defend their food because there is so much of it that they will invariably eat something. This asymmetry in motivation leads owners to give up some food (play Dove) rather than defend it all (play Hawk). So yes, harassment is a specific behavioral mechanism by which beggars impose costs on owners, but the general logic comes from the asymmetrical Hawk-Dove game.

199: here it is claimed that there was clear possession during "collect near" - how exactly did this manifest itself?

257, 262: I don't think the "however" is warranted here since the no-outliers analysis shows essentially the same result as the full model? further, in 263 i would remove the word "significant" from "significant trend" (an oxymoron)

331: "same mass-specific energy demands as adults" - really? I always thought that metabolic rate scales *allo*metrically with size (not isometrically), i.e. adults would have lower metabolic rate per unit weight.

·

Basic reporting

The writing, in general, is clear and understandable. The introduction is thorough in its’ background and references. The figures are relevant and explanatory. The raw data is provided. The following should be corrected:

Lines 51-53: Please cut “Both models Theoretical models of begging have focussed on the nutritional demands of offspring and food solicitation interactions with parents (Grafen 1990; Zahavi 1975).” It is a repeated section.

Line 53: “Both models…” Would be helpful to specify what the two models are.

Line 55: Should be “Mock and Parker 1997)”

Lines 53-58: “Both models…” This sentence is confusing and should be rewritten for clarity.

Line 63: Cut “in nature”

Lines 74-80: Should be “Food sharing is commonly observed in nonhuman animals, including birds (termed “allofeeding”, Evans and Marler 1994; Heinrich 1988; Stacey and Koenig 1990; Thiollay 1991), insects (Bolten et al. 1983; Boggs 1995; Vahed 1998), fish (Griffiths and Armstrong 2002),and mammals, including vampire bats (Carter and Wilkinson 2013; Denault and McFarlane 1995; Wilkinson 1984), cetaceans (Johnson 1982; Hoelzel 1991) mice (Porter, Moore and White 1981), spotted hyenas (Holekamp and Smale 1990), lions (Cooper 1991) and wolves (Dale et al. 2017).”

Line 137: It would be helpful to have a description of what makes a food item more or less difficult to exploit.

Line 210: Should be: “per otter and trial were fitted and were modelled…”

Figure 1: Is it possible to include r for the regression line on the chart?

Experimental design

It is an interesting research question in line with the depth of research conducted on begging and sharing. For the most part, the methods are clear and replicable. The statistical analyses are thorough and the r code is provided. The animals were treated well within ethical standards and the research was approved by the university ethics committee and both zoos. Please address the following:

Lines 174-175: Why weren’t the levels of exploitability randomized? By the third repetition, could it have been possible for the otters to predict and might this have influenced their behavior?

Line 185: “within arm’s reach” The otter's arm's reach? 1m is a very long arm for an ASCO. Or is this a human's arm's reach? Please clarify.

Results: Who were the juveniles and adults begging and sharing from? Were juveniles begging from adults and adults sharing with juveniles or the opposite? Was it more within categories? Juveniles sharing and begging with juveniles and the same with adults? Is it possible to include this in the analysis?

Validity of the findings

The data and r code are included and the statistical analysis is sound. The conclusions are clear and come out of the research question and previous research. One possible aspect to discuss: It is surprising that there was no difference between the behavior in the animals from the two different zoos based on the differences in group sizes, differences in enclosures, and feeding schedules and provided foods. What might have accounted for that?

Additional comments

Overall, this is a fine study on an interesting question. There is thorough background research and, in general, it is well written. The statistical analysis is valid and the results and conclusions are interesting. Please address the provided comments.

---

## Round 0.2 · Minor Revisions

Thank you very much for your detailed and thoughtful response to the extensive reviewer feedback from the last round of reviews. I have just a few very tiny requests before I can formally accept the MS.

Line 88: place a comma after “in this study.” Check for other clauses that should be followed by commas.

Also, use commas after e.g. and i.e. (e.g., line 136, 138). Fix throughout.

Should line 198 just be less than 1m OR within an otter’s arm reach, because if the two are very different lengths, as per Preston Foerder’s comment in the last review, this correction still seems inadequate. Why was not otter’s arm length used consistently?

Why is there no standard error bar on the trout high abundance condition in Figure 2? If no variability here, then fine as is.

---

## Round 0.3 · accepted · Accept

Thank you for these additional minor edits. I agree with presenting the data in its raw form for figures. Congratulations on a nice contribution.